# Metabolic Reprogramming of Urothelial Carcinoma—A Theragnostic Target for Betulinic Acid

**DOI:** 10.3390/ijms26125598

**Published:** 2025-06-11

**Authors:** Anirban Ganguly, Aratrika Halder, Keara Healy, Stephanie Daugherty, Shingo Kimura, Rajkumar Banerjee, Jonathan M. Beckel, Pradeep Tyagi

**Affiliations:** 1Department of Pharmacology and Chemical Biology, University of Pittsburgh School of Medicine, Pittsburgh, PA 15213, USA; ang334@pitt.edu (A.G.); keh208@pitt.edu (K.H.); slz8@pitt.edu (S.D.); shk361@pitt.edu (S.K.); jmbeckel@pitt.edu (J.M.B.); 2Department of Urology, University of Pittsburgh School of Medicine, University of Pittsburgh, Pittsburgh, PA 15213, USA; 3Oils, Lipids Science & Technology Division, CSIR-Indian Institute of Chemical Technology, Hyderabad 500007, Indiabanerjee@iict.res.in (R.B.)

**Keywords:** betulinic acid, mitomycin C, metabolic reprogramming, urothelial carcinoma, theragnostic

## Abstract

A pivotal role of metabolic reprogramming in urothelial carcinoma is hallmarked by the dependence of two-fold faster proliferation of urothelial carcinoma cell line T24 than benign cell line TRT-HU1 on five-fold higher glucose (basal) 16 mM vs. 3 mM in McCoy’s 5A media and Keratinocyte Serum Free media, respectively. Here, we report that an additional 10% increase to 17.6 mM and 3.3 mM glucose significantly shortens the doubling time by 3 h and 1 h for T24 and TRT-HUI, respectively. T24 grown at 17.6 mM glucose lowers the confocal localization of the fatty acid mimetic, Betulinic Acid (BA) conjugated to FITC (BA-FITC) with Mito Tracker Red (mitochondrial marker), which doubles the IC50 of BA and BA-FITC by lowering cell cycle arrest in the G0/G1 phase from 54.2% to 43.8% and caspase-3/7 mediated apoptosis and by reversing *caspase-3*, *p53*, *PTEN*, *GAPDH,* and *XIAP* gene expression induced by BA in T24 grown at basal glucose (16 mM). Besides slowing the glycogen and pH decline of T24 at basal glucose, BA exhibited an eight-fold higher IC50 than Mitomycin C (MC) on TRT-HU1 by not mimicking the glucose-insensitive cycle arrest and apoptosis of MC. Overall, the glucose sensitivity of the lower IC50 of BA-FITC and BA on T24 vs. TRT-HU1 supports the safety of BA conjugates for theragnostic purposes.

## 1. Introduction

The pivotal role of metabolic reprogramming in the onset and rapid recurrence of urothelial carcinoma [1,2,3] is affirmed by the genomics of resected tumor and by metabolomics of the patient’s urine. The overexpression of genes driving glycolysis [1,2,3,4,5] and fatty acid metabolism in tumors [6] is consistent with the urinary detection of 12–19 metabolites [7] and glycogen depletion [8] coupled with fatty acid accumulation in tumor cells shed into urine by patients, converse of the metabolomics for normal urothelial cells shed into the urine.

(i) State-of-the-art: The multi-omics of tumors resected from bladder cancer patients are attested by the dependence of two-fold faster proliferation [9] of urothelial carcinoma cell line T24 relative to benign urothelial cell line TRT-HU1 on five-fold higher (basal) glucose levels of 16 mM and 3 mM in McCoy’s 5A media and Keratinocyte Serum Free media, respectively (Figure 1). A simple correspondence between human blood glucose levels and glucose levels in commercially available growth media for neoplastic and non-neoplastic cell lines raises the question about the appropriate increase in glucose levels for studying urothelial carcinoma biology with the T24 cell line: whether it should be a 56% increase from basal levels of 16 mM to 25 mM [10] or to 17.6 mM, as used here. The sparse discussion of this topic in the published literature highlights the following (ii) gaps in the literature: The glucose levels of 3 mM [52.5 mg/100 mL (dL)] in Keratinocyte Serum Free media for non-neoplastic cell line TRT-HU1 fall below the lower threshold for the normal human blood glucose range of 3.9–7 mM (<120 mg/dL) even with a 10% increase. Whereas even without the additional 10% increase in glucose, the basal glucose levels of 16 mM (280 mg/dL) in commercially available McCoy’s 5A media for neoplastic cell line T24 correspond to the sustained hyperglycemia of a severely diabetic person hallmarked by glycosylated hemoglobin levels, HbA1C ≥ 11. Does faster proliferation of T24 at glucose levels of a diabetic person foreshadow the association of significantly higher recurrence rate and worse recurrence-free survival of non-muscle invasive bladder cancer patients having HbA1C ≥ 7 in a retrospective study [11] and a higher risk of urothelial carcinoma in diabetic patients [12,13]? Hyperglycemia is also attributed as a factor in the lower uptake of 2-fluoro-2-deoxy-D-glucose (2-FDG)—a glucose surrogate—in positron emission tomography (PET) of tumors [14,15] and in the lower sensitivity of antibody drug conjugates, Enfortumab vedotin [16,17]. Does the need for >5-fold higher glucose for two-fold faster proliferation of neoplastic cell lines than non-neoplastic cell lines embody the role of metabolic reprogramming in urothelial carcinoma? Does the growth of the T24 cell line at glucose levels of a diabetic person underestimate the in vivo cytotoxic effect of natural products in animals and patients having normal blood glucose levels?

Betulinic Acid (BA) is one of the many active ingredients [18] in the anti-inflammatory Oleogel S-10 approved recently in Europe for the treatment of dystrophic epidermolysis bullosa [19]. In addition, the anticancer activity of topical BA on horse melanoma [20] merited the registration of a pilot clinical study (NCT00346502), and urothelial carcinoma is also amenable to topical treatment [21]. Past research on BA highlighted the induction of apoptosis via caspase-dependent cell death, depolarization of mitochondrial membrane potential [22,23,24,25,26], and cell cycle arrest [26,27,28,29]. While prior research supports the premise of BA conjugates [22] as a theragnostic, the role of metabolic reprogramming in the elevated uptake of BA by neoplastic cell lines compared to non-neoplastic cell lines is not known [22].

(iii) Research question: Past research makes it abundantly clear that neoplastic cell lines proliferate faster at glucose levels that are greater than two-fold higher than normal blood glucose levels [30], whereas non-neoplastic cell lines (TRT-HU1) begin to exhibit signs of osmotic stress at higher glucose levels [7]. Accordingly, we decided to raise the glucose in growth media by 10% from basal levels of 16 mM and 3 mM for neoplastic and non-neoplastic cell lines, respectively. We hypothesized that if rapid proliferation of urothelial carcinoma is fueled by fatty acid synthesis and oxidation, then a 10% rise in glucose would decrease the intracellular accumulation of the fatty acid mimetic, BA, as well as oppose the pharmacological effect of BA on proliferation, cell cycle arrest, and apoptosis.

To test our hypothesis, we chemically conjugated BA [22] with fluorescein isothiocyanate (FITC) (Appendix A) for determining the confocal co-localization of BA-FITC with Mito Tracker Red-labeled mitochondria in T24 and TRT-HU1 cell lines of human urothelium. We determined the cytotoxicity and safety of BA relative to the standard chemotherapeutic drug, Mitomycin C (MC) [21], on T24 and TRT-HU1 cell lines grown in McCoy’s 5A media and Keratinocyte Serum Free media containing 16 mM and 3 mM glucose (basal), respectively, and a 10% increase from respective basal levels. The cytotoxic effect of MC and BA on two other cell lines is shown in the Appendix A. We compared the effect of BA and MC on cell cycle arrest and caspase-3/7-mediated apoptosis [27] and only investigated the glucose-sensitive cytotoxic effect of BA by determining the effect of BA on the expression of genes involved in apoptosis [23,24,26,27,28], glycogenolysis [1,2,3,4,5,8], and glycolysis, causing the extracellular acidification amidst glucose scarcity [31] (Figure 1).

## 2. Results

### 2.1. Effect of 10% Glucose Increase on Doubling Time of Cell Lines

The growth of urothelial carcinoma cell line T24 in McCoy’s 5A media containing 16 mM of glucose reproduced the previously reported faster doubling time of 19 h [9], which is more than two-fold faster than the 40 h doubling time of non-neoplastic human bladder cell lines TRT-HU1 grown in Keratinocyte Serum Free media containing 3 mM glucose (Table 1).

It is self-evident that the two-fold faster proliferation of T24 is fueled by the Warburg effect, dependent on >five-fold higher glucose than TRT-HU1 (Figure 1) to sustain a higher rate of glycolysis. The role of the Warburg effect in faster T24 proliferation is also attested by a reduction in the doubling time from 18.3 ± 0.048 h to 15.5 ± 0.126 h brought about by a mere 10% rise in glucose levels from 16 mM to 17.6 mM, whereas a proportional 10% increase in glucose from 3 mM to 3.3 mM for TRT-HU1 only shortened the doubling time from 40.7 ± 0.024 h to 39.4 ± 0.072 h.

### 2.2. Effect of 10% Glucose Increase from Basal Levels on MC-Evoked Cytotoxicity, Cell Cycle Arrest, and Apoptosis

MTT results plotted in Figure 2 display comparable cytotoxicity of MC on normal TRT-HU1 and neoplastic urothelial cancer cell T24 with an IC_50_ of 3.7–4.3 µM at basal glucose (Figure 2A), and a 10% increase in glucose from basal levels did not elicit any change in IC_50_. The glucose-insensitive, cytotoxic actions of MC are also confirmed by comparable induction of caspase-3/7 apoptotic activity (Figure 2B) and non-selective cell cycle arrest in the G0/G1 phase (Figure 2C) of T24 and TRT-HU1 cells at basal glucose and at a 10% increase in glucose from basal levels of 16 mM and 3 mM, respectively. Cell cycle arrest was determined after cell synchronization by double thymidine block of 18 h followed by continuous exposure to MC for 12 h.

### 2.3. Effect of 10% Glucose Increase from Basal Levels on BA-Evoked Cytotoxicity, Cell Cycle Arrest, and Apoptosis

Unlike MC, BA evoked concentration-dependent higher cytotoxicity in T24 cells than in TRT-HU1 cells, and the BA IC_50_ on T24 was significantly lower (17.9 ± 0.268 µM) (Figure 3A) than the IC_50_ (34.5 ± 1.165 µM; *p* ≤ 0.01) on TRT-HU1 grown at respective basal levels of glucose. The difference in IC_50_ is suggestive of relative resistance of normal urothelial cells to BA and potentially improved safety profile of BA [14] as a potential theragnostic. A rightward shift in BA IC_50_ from 18.3 ± 0.048 μM to 35.3 ± 0.386 μM in T24 cells grown at 10% increased glucose from basal levels was associated with a significant reduction (*p* ≤ 0.05) in caspase-3/7 apoptosis (Figure 3B) and a markedly lower percentage of cell cycle arrest (Figure 3C). While BA arrested a higher percentage of T24 cells than TRT-HUI (54.2% vs. 41.5%) at basal glucose, a 10% increase in glucose from basal levels brought parity, 43.8% vs. 46.8%.

### 2.4. Combined and Isolated Effect of BA and 10% Glucose Increase on Gene Expression

While glucose-insensitive anticancer actions of MC are consistent with published findings, glucose-sensitive anticancer actions of BA warranted further investigations without using MC as a control. The increased caspase-dependent apoptosis of BA-treated T24 cells at basal glucose (Figure 3B) was accompanied by the up-regulation of *Caspase-3*, *p53, Phosphatase and Tensin Homolog deleted on Chromosome 10 (PTEN), and Glyceraldehyde-3-Phosphate Dehydrogenase (GAPDH)*, together with the downregulation of the anti-apoptotic gene, *X-linked inhibitor of apoptosis protein (XIAP)*, at basal glucose (Figure 4A). Because BA-induced upregulation of gene expression in T24 cells grown at basal glucose is reversible with a 10% increase in glucose from basal levels (Figure 4B), we inferred that BA targets glucose scarcity-induced metabolic reprogramming of T24 cells. Meanwhile, the expression of these genes in TRT-HU1 was not altered by a 10% increase in glucose from basal levels (Figure 4C,D).

### 2.5. Effect of 10% Glucose Increase on Mitochondrial Localization and Toxicity of Betulinic Acid Conjugated to FITC

We examined the mitochondrial localization of Betulinic Acid (BA) conjugated to FITC (BA-FITC) (Figure 5) [22] by confocal microscopy and compared the colocalization of green fluorescence emitted by BA-FITC with the red fluorescence of Mito Tracker (mitochondrial marker) in T24 and TRT-HU1 cells, and red fluorescence was also examined independent of BA-FITC (Appendix A). BA-FITC exhibited higher intracellular uptake and mitochondrial localization in T24 relative to TRT-HU1 grown at basal glucose levels. The mitochondria as a site for the antiproliferative action of BA is affirmed by the decline in Pearson’s correlation coefficient (Appendix A) from 0.91 to 0.75 for the colocalization in T24 cells grown at a basal level of 16 mM (Figure 5A) and at 17.6 mM (Figure 5B), respectively. Lower mitochondrial localization of BA-FITC at higher glucose resulted in more than two-fold higher IC_50_ of BA-FITC from 13.5 ± 0.054 µM (Figure 5C) to 31.4 ± 0.324 µM (Figure 5D) for T24 cells, whereas the increase in IC_50_ for TRT-HU1 was relatively minor compared to T24. While increased mitochondrial accumulation of BA and BA-FITC in T24 at basal glucose aligns with the caspase-dependent apoptosis (Figure 3B) and cell cycle arrest (Figure 3C), reduced mitochondrial accumulation of BA-FITC in T24 cells grown at a 10% increase in glucose replicates a cell phenotype that is resistant to antibody drug conjugates, Enfortumab vedotin [16,17].

The intensity of Mito Tracker Red fluorescence in neoplastic (T24) and non-neoplastic cells (TRT-HU1) left untreated with BA-FITC (Appendix A) was identical to the cells that were treated sequentially with Mito Tracker Red and BA-FITC (Figure 5). Mito Tracker Red (Appendix A) also mirrored the relative reduction and rise from basal glucose in the mitochondrial uptake of BA-FITC by T24 and TRT-HU1 cells, respectively, grown at 10% higher glucose (Figure 5). A discernible random variation in T24 cell viability (%) at the lowest dose of BA and BA-FITC is fully accounted for, by the fitting of a sigmoidal dose–response curve to a non-linear regression model by GraphPad Prism for determining IC_50_, a mean response that is 50% higher than the response at lowest dose [32]. As reported elsewhere [33], the variation in the viability percentage at the lowest dose only makes a marginal difference to IC_50_ and a 4 μM difference between the IC_50_ of BA and BA-FITC remains even when the viability percentages are normalized to the cell viability at the lowest dose of BA and BA-FITC instead of viability in the vehicle group (Appendix A).

### 2.6. Effect of BA on Glycogen and pH Decline Amidst Glucose Scarcity

Here, we checked whether the glycogen depletion found in tumor cells shed into urine by patients [8] is reproduced by T24 cells exposed briefly to glucose scarcity after growing for 8 h at basal glucose (Figure 6A and Figure 7A) or at 10% increased glucose from basal levels (Figure 6B and Figure 7B). Importantly, the 8 h time point is the midpoint for the doubling time at 10% increased glucose (Table 1). We created glucose-scarce conditions for glycogen assay and for measuring the acidification of extracellular space by replacing the buffered growth media of T24 and TRT-HU1 cells with an unbuffered isotonic solution of pH 7.4. The lack of a temporal change in pH of isotonic unbuffered solution at incubation conditions of 37 °C and 5% CO_2_ for the assay period simplifies the interpretation of findings. Relatively faster proliferation (Table 1) of T24 cells at 10% increased glucose resulted in a steeper decline of glycogen content (Figure 6B) and of pH (Figure 7B) relative to TRT-HU1 cells. T24 cells grown at basal and at 10% higher glucose mimicked the glycogen decline reported in patients’ tumor cells shed into urine [8].

A slower decline of glycogen (Figure 6C,D) and of pH (Figure 7C,D) in BA-treated cells is interpreted as BA slowing the proliferation of T24 cells amidst glucose scarcity by BA-evoked cell cycle arrest (Figure 3B) and mitochondrial localization of BA (Figure 5A,B)—fatty acid mimetic—inhibiting fatty acid synthesis from glycogen to fuel the faster T24 proliferation amidst glucose scarcity.

## 3. Discussion

The in vitro evidence described here is sufficient to reject the null hypothesis of glucose elevation in growth media not impacting the elevated cytotoxicity of Betulinic Acid in neoplastic cell lines compared to non-neoplastic cell lines. The evidence makes it abundantly clear that a critical determinant for cancer cell selectivity of Betulinic Acid is metabolic reprogramming of cancer, which enables T24 cells to proliferate amidst glucose scarcity. A pivotal role of metabolic reprogramming in carcinogenesis [1,2,3,4,5] is substantiated by a 10% increase in glucose from a basal level of 16 mM, decreasing the doubling time of the T24 cell line and abolishing the elevated cytotoxicity of BA in T24 cells by reducing the internalization of BA-FITC into mitochondria. T24 cells treated with BA arrest the decline of glycogen and pH amidst glucose scarcity. BA-FITC mirrored the glucose-sensitive mitochondrial localization of Mito Tracker Red in neoplastic and non-neoplastic cell lines grown at a 10% increase in glucose. Compared to the localization and viability at basal glucose levels, the decline in mitochondrial localization of Mito Tracker Red and BA-FITC in neoplastic cell lines grown in 10% increased glucose contrasts with the rise in mitochondrial localization of Mito Tracker Red and BA-FITC in non-neoplastic cell lines cultured at 10% increased glucose from basal levels.

The contrasting effect of a 10% glucose rise on mitochondrial localization of Mito Tracker Red and BA-FITC in neoplastic and non-neoplastic cell lines implies that BA-FITC could be relying on mitochondrial membrane transformations for cancer selectivity [34]. Moreover, the mitochondrial membrane transformations of neoplastic cell lines could be replicated by non-neoplastic cell lines grown at 10% increased glucose in growth media, which attenuates the cytotoxicity of Betulinic Acid on non-neoplastic urothelial cells (TRT-HU1 and HBDEC). In contrast, a 10% rise in glucose made the neoplastic cell lines (T24 and RT4) resistant to the anticancer effect of Betulinic Acid. T24 cells grown at 10% increased glucose not only successfully resisted the mitochondrial localization of BA-FITC but also reversed the BA-induced gene expression [27]. Accordingly, we inferred that the lower IC_50_ of BA as well as BA-FITC in T24 cells [24] is dependent on glucose scarcity-driven upregulation of pro-apoptotic genes *Caspase-3*, *p53*, and *PTEN* and the downregulation of anti-apoptotic *XIAP* [35] gene along with metabolic marker *GAPDH*. The caspase-3/7 apoptotic assay substantiated that a 10% increase in glucose reverses the *Caspase-3* upregulation induced by BA at basal glucose.

The premise of BA conjugates as diagnostic and therapeutic agents for urothelial carcinoma is based on the lower IC_50_ of BA-FITC than of BA on T24 cells. This premise is strengthened by a prior report of higher cytotoxicity of BA-conjugated bis-arylidene oxindole than of unconjugated BA on a host of cell lines [22]. The localization of BA-FITC in mitochondria of T24 cells is consistent with the past reports on BA depolarizing the mitochondrial potential of cancer cells [27,36,37] and inhibiting topoisomerase 1B, involved in mitochondrial translation and carcinogenesis [28]. A lower mitochondrial localization of BA-FITC in T24 grown at 17.6 mM glucose resulted in a two-fold higher IC_50_ than the IC_50_ of BA-FITC measured in T24 cells grown at 16 mM glucose. A dramatic rise in IC_50_ of BA and BA-FITC on T24 cells grown at 17.6 mM glucose mimics the lower sensitivity of diabetic bladder cancer patients to antibody drug conjugates, Enfortumab vedotin [16]. The glucose-sensitive localization of BA-FITC, a fatty acid mimetic, in the mitochondria of T24 cells corroborates fatty acid accumulation [8,38,39] seen in cancer cells shed into the urine of urothelial carcinoma patients. The coincidence of fatty acid accumulation and glycogen depletion in cancer cells [8] aligns with the urine metabolomics and aberrant expression of glycolysis genes in resected tumors [7]. The glycogen depletion [8] seen in urine of bladder cancer patients and in T24 cells likely reflects glycogenolysis driving fatty acid synthesis, decreased glycogenesis [40], and oxidative phosphorylation [31]. Accordingly, we reason that cancer cells internalizing Betulinic Acid, a fatty acid mimetic, are likely to inhibit enzymes involved in de novo synthesis of fatty acids by the pentose phosphate pathway amidst glucose scarcity, which warrants further investigation in future studies.

Because BA was inferior to MC in inducing an antiproliferative effect on muscle-invasive cancer cell line T24 and non-muscle-invasive cancer cell line RT4 (Appendix A), we can conclude with reasonable confidence that BA is also likely to be inferior to cisplatin on muscle-invasive and non-muscle-invasive cancer cell lines. While the mechanism of action and gene expression involved in the anticancer effect of MC are well known, the same is not the case for BA. Therefore, we probed whether BA selectively targets metabolic reprogramming of urothelial carcinoma cell line T24, and the absence of the same metabolic reprogramming in non-neoplastic urothelial cells, TRT-HU1, results in minimal harm [24]. Based on our findings, T24 grown in McCoy’s 5A media may be appropriate for screening drugs that intercalate with DNA in a glucose-insensitive manner like Mitomycin [41], but T24 cells may be underestimating the in vivo efficacy of anticancer drugs such as Betulinic Acid that target metabolic reprogramming of cancer [42] with glucose-sensitive mitochondrial localization.

The glucose-sensitive mitochondrial localization of BA is deterministic in BA-evoked cell cycle arrest and caspase-3/7-mediated apoptosis [18,23,26]. We analyzed the effect of MC and BA on the cell cycle after cell synchronization via double thymidine block, which arrests the cells at the G1 phase. Upon release of the thymidine block, the G0/G1 phase lasts 9–10 h, nearly the midpoint of the doubling time of 18 h for T24 cells. The caveats of doubling time and cell synchronization are relevant in interpreting BA-evoked cell cycle arrest in the G0/G1 phase as opposed to arrest in the S or G2/M phase [28] seen in other cancer cell lines hallmarked by different doubling times. As opposed to the arrest of nearly an identical percentage of T24 and TRT-HU1 cells by MC at basal and at 10% increased glucose, BA arrested a higher percentage of T24 cells grown at basal glucose than TRT-HU1 cells in G0/G1 phase.

To assess the effect of BA on glycogen and pH decline amidst glucose scarcity, we replaced the buffered growth media containing 16 mM or 17.6 mM glucose with unbuffered isotonic solution at pH 7.4 after cells had reached their log growth phase after having been plated for 8 h. The absence of the buffering effect from the bicarbonate buffer in the isotonic solution enables the detection of pH decline ensuing from rapid proliferation amidst glucose scarcity. BA evoked cell cycle arrest [26,43,44,45] in 54.2% of T24 cells grown at basal glucose, sheds light on the slower decline in pH and glycogen content amidst glucose-scarce conditions for 3 h [46]. Therefore, innately fast proliferation of T24 cells amidst glucose scarcity enhances lactic acid production, raising the assembly and export of acid equivalents to acidify the extracellular space [47]. Thus, “Warburg effect” [31,38,48] fuels T24 cell proliferation at basal glucose levels of 16 mM, hallmarked by glycogenolysis [40], accelerated glycolysis, faster formation of pyruvate [38], and lactate conversion [38,39], which causes acidification of the extracellular space [47]. In recent studies, hyperglycemia triggering lactate overproduction was identified as a cause for lower sensitivity of Enfortumab vedotin [16] in bladder cancer patients, and acidic urine was associated with increased recurrence of bladder cancer post-BCG treatment [49]. Importantly, higher lactate production by cancer cells is the rationale underpinning the tumor detection by pH-sensitive magnetic resonance imaging (MRI) probes [50,51] and by pH-sensitive fluorescent probes [52].

Moreover, our findings raise the following question: what is the clinically relevant rise in basal glucose to mimic a large majority of bladder cancer patients with normal blood glucose? whether T24 cells should be studied at 56% higher than basal glucose levels of McCoy’s 5A media [10] or a 10% increase in glucose adopted here. Future studies can employ a Seahorse metabolic analyzer [53] to confirm the dependence of higher lactate production on faster glycolytic rate of T24 cells than TRT-HU1 cells and oxygen consumption [31]. A faster proliferation of T24 cells with a 10% increase from basal glucose corroborates the higher risk of urothelial carcinoma in diabetic patients [12,54] and the arrest of carcinogenesis by drugs that inhibit glycolysis [1,2,3,4,5]. We could also draw an intriguing parallel between the decline in the mitochondrial localization of FITC-tagged BA in T24 cells grown at 10% increased glucose and reduced tumor uptake of 2-FDG in PET scans performed without lowering the blood glucose levels [14]. Betulinic Acid conjugated radionuclides or dyes could be used in the diagnosis and treatment of bladder cancer. We posit that any hypothetical impairment of Betulinic Acid conjugates by hyperglycemia can be averted by combining BA with newer glucagon-like peptide-1 receptor (GLP-1R) agonists [55]. Likewise, the influence of hyperglycemia in PET scans of diabetic cancer patients [56] can be mitigated by the acceleration of glucosuria [57] with the use of sodium-glucose co-transporter-2 (SGLT-2) inhibitors, which may incur hypoglycemia risk. Retrospective studies on diabetic bladder cancer patients treated with SGLT-2 inhibitors [58] can answer whether the high risk of urothelial carcinoma in diabetic patients stems from hyperglycemia or glucosuria. Interestingly, BA-induced *GAPDH* upregulation in T24 cancer cells [9] may offer a potential alternative mechanism for raising FDG uptake into tumors without triggering the hypoglycemia risk [14]. Overall, our findings demonstrate that the anticancer action of BA is dependent on metabolic reprogramming of T24 cancer cells.

## 4. Materials and Methods

### 4.1. Materials

Betulinic Acid (Catalog No. B8936; 456.7 Daltons), Mitomycin C (Catalog No. M4287; 334.3 Daltons), and fluorescein isothiocyanate (FITC) (Catalog No. 46950; 389.38 Daltons) were procured from Sigma Aldrich, St. Louis, MO, USA, whereas Mito Tracker Red (Catalog No. M22425) was procured from Invitrogen, Carlsbad, CA, USA. The conjugation of Betulinic Acid with FITC was confirmed by the rise in molecular weight (Appendix A). TRT-HU1 (Catalog No. CVCL_M720) and Keratinocyte Serum Free media (Catalog No. 10724011) were procured from Cellosaurus, Lausanne, Switzerland, and Gibco Thermo Fisher Scientific, Waltham, MA, USA, respectively, whereas T24 (Catalog No. HTB-4) and McCoy’s 5A media (Catalog No. 16600082) were purchased from ATCC, Manassas, VA, USA, and Gibco Thermo Fisher Scientific, Waltham, MA, USA, respectively. Thymidine (Catalog No. T1895) and Propidium Iodide (Catalog No. 81845) for cell synchronization in cell cycle analysis were procured from Sigma-Aldrich, St. Louis, MO, USA. Caspase3/7 assay kit including Camptothecin (Catalog No. C10430) was procured from Invitrogen, Carlsbad, CA, USA.

### 4.2. Cell Culture

Urothelial carcinoma T24 cells and non-cancerous human urothelial TRT-HU1 were grown in respective media supplemented with 10% fetal bovine serum (FBS) at 37 °C in a humidified 5% CO_2_ atmosphere. TRT-HU1-KSFM Medium was supplemented with Epidermal Growth Factor (EGF) and Bovine Pituitary Extract (Invitrogen). Cells were used within 5 passages after initial thawing. Basal and 10% increased glucose levels in media for T24 cells were 16 mM and 17.6 mM, respectively, whereas basal and 10% increased glucose for TRT-HU1 were 3 mM and 3.3 mM, respectively. Doubling time was measured after the first two passages using the formula [T × (ln2)]/[ln (Xe,Xb)], where T is time in hours, Xe is the number of cells at the end, and Xb is the number of cells in the beginning. A hemocytometer was used for cell counting.

### 4.3. Cell Viability Assay (MTT)

After two passages, 5000 cells (T24 and TRT-HU1) were plated in each well of 96-well plates and incubated overnight before 18 h exposure to increasing concentrations of BA or MC, and the concentration reducing cell viability by 50% (IC50) was determined by plotting MTT assay readings taken at 570 nm using a TECAN Spark 20 M spectrophotometer, TECAN Life Sciences, Männedorf, canton of Zürich, Switzerland. The plotted cell viability percentages are normalized viability percentages relative to the vehicle group.

### 4.4. Cell Cycle Analysis After Cell Synchronization by Double Thymidine Block

T24 and TRT-HU1 cells were grown in T25 flasks in basal and at 10% increased glucose from basal until confluency. Both the cell lines were sub cultured in 1:3 flasks and incubated overnight at 37 °C and then subjected to a double thymidine (2 mM) block for 18 h each. After the first block, cells were washed with 1× PBS after removing the thymidine and then allowed to grow for 9 h on fresh pre-warmed media (basal and 10% increased) before the second thymidine block for 18 h. After again removing thymidine, cells were treated at respective IC50s of BA (20 µM) and MC (4 µM) for 12 h and analyzed by the propidium iodide method using a BD Accuri C6 plus flow cytometer, BD Biosciences, Franklin Lakes, NJ, USA.

### 4.5. Caspase-3/7 Apoptotic Assay

Cells were plated as described in the MTT assay. Cells were then either left untreated or treated with BA (20 µM) and MC (4 µM) for 12 h using Camptothecin (5 µM) as a positive control. The caspase-3/7 assay was performed according to the manufacturer’s protocol, and the fluorescence intensity was measured at an excitation/emission wavelength of 590/610 nm.

### 4.6. Quantitative PCR

Untreated T24 and TRT-HU1 cells or cells treated with BA (20 µM) for 12 h were washed with 1× PBS before lysing with 1 mL Trizol (Invitrogen, Carlsbad, CA, USA) for total RNA extraction, as per manufacturer’s instructions. RNA purity and concentration were determined using nanodrop 2000 (Thermofisher Scientific, Waltham, MA, USA). cDNA was created using the High-Capacity cDNA Reverse Transcription Kit from Applied Biosystems, Thermo Fisher Scientific, Waltham, MA, USA, with 1 μg of total RNA. PCR was performed with a HotStarTaq Master Mix Kit (Qiagen, Valencia, CA, USA) using sequence-specific primers for the Glyceraldehyde-3-Phosphate Dehydrogenase (GAPDH), Caspase-3, p53, Phosphatase and Tensin Homolog deleted on Chromosome 10 (PTEN), and X-linked inhibitor of apoptosis protein (XIAP), designed in-house using previously published sequences on online primer design tool of Primer3 https://primer3.ut.ee/ (accessed on 20 May 2024)). The primers were as follows: GAPDH, Caspase-3 (NM_004346.4) L: 5′-ACTGGACTGTGGCATTGAGA-3′; R: 5′-GCACAAAGCGACTGGATGAA-3′, p53 L: 5′-TGGCCATCTACAAGCAGTCA-3′; R: 5′-GGTACAGTCAGAGCCAACCT-3′, PTEN (NM_000314.8) L: 5′-ACCGGCAGCATCAAATGTTT-3′; R: 5′-AGTTCCACCCCTTCCATCTG-3′, XIAP (NM_001167.4) L: 5′-TGCTCACCTAACCCCAAGAG-3′; R: TCCGGCCCAAAACAAAGAAG-3′. Quantitative PCR (CFX Connect Real-Time system, BIORAD, Hercules, CA, USA) was performed as per the manufacturer’s instructions.

### 4.7. Confocal Microscopy

Pre-heated 18 mm coverslips (Warner’s instruments, Holliston, MA, USA) were placed in each well of 6-well plates before plating 1.2 × 10^6^ cells/well and allowed to grow overnight before 2 h exposure to media with or without 20 µM BA-FITC. Cells were then washed with 1× PBS before 15 min treatment with 25 nM Mito Tracker Red (Invitrogen, Carlsbad, CA, USA). The cells were again washed with 1× PBS before mounting coverslips on slides under a confocal microscope (Olympus Fluoview FV1000, Olympus Global, Shinjuku City, Tokyo, Japan). Z-stack images were taken at 60× magnification with Laser HV454v, 2× gain, and 13% offset.

### 4.8. Glycogen and pH Assay

In this study, 5000 T24 and TRT-HU1 cells were plated in each well of 96-well plates and allowed to grow for 8 h in respective growth media with basal or 10% increased glucose, with and without BA treatment at 37 °C and 5% CO_2_. Media was subsequently replaced with an unbuffered, isotonic solution of pH 7.4 to create glucose scarcity for glycogen assay by fluorometric glycogen assay kit ab65620 (Abcam, Cambridge, MA, USA) as per the manufacturer’s instructions using glycogen standard and excitation/emission of 535/587 nm at 30, 60, and 120 min, respectively. Cell proliferation amidst scarcity determined the rate of lactic acid production and pH decline from an initial pH of 7.4, which was measured by an Accumet 950 pH/ion meter (Thermofisher Scientific, Waltham, MA, USA) at 0, 30, 60, 120, and 180 min.

### 4.9. Statistical Analysis

Data are expressed as mean ± SD. Time-dependent decline in glycogen and pH of cell lines relative to isotonic solution of pH 7.4 was plotted, and significance was analyzed by two-way ANOVA using GraphPad Prism version 10 software. Pearson’s coefficient was determined for colocalization, and significant difference was assessed by Student’s *t*-test. Results were considered significant at *p* < 0.05.

## 5. Conclusions

BA differs from MC in targeting glucose scarcity-induced metabolic reprogramming of urothelial carcinoma for lower toxicity on normal human urothelial cells. While traditional safe human use of birch bark as a folklore medicine is authenticated by an eight-fold higher IC50 of BA on TRT-HU1 than Mitomycin C, BA-FITC also mimicked FDG in metabolic reprogramming-dependent entry into T24 cancer cells. Overall, findings hint at the promise of BA conjugates as a next-generation theragnostic for urothelial carcinoma.

## Figures and Tables

**Figure 1 ijms-26-05598-f001:**
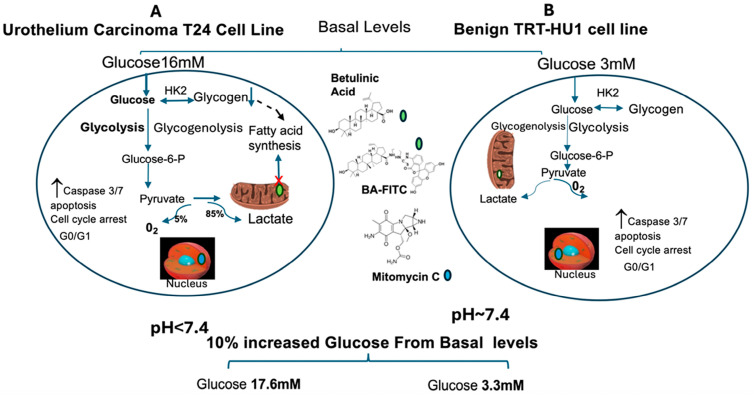
Two-fold faster proliferation of the urothelial carcinoma (T24) cell line relative to benign urothelial cells (TRT-HU1) is fueled by the “Warburg effect”, which is sustained by >5-fold higher 16 mM glucose in McCoy’s 5A medium (**A**) relative to 3 mM glucose in Keratinocyte Serum Free media for TRT-HU1 (**B**). A 10% increase from basal levels to 17.6 mM and 3.3 mM for T24 and TRT-HU1 cell lines shortens the doubling time by 3 h and 1 h, respectively, and lowers the confocal co-localization of natural fatty acid mimetic, Betulinic Acid (BA-FITC), with Mito Tracker Red (mitochondrial marker), which coincides with slower decline of glycogen stores and of pH amidst glucose scarcity. Findings highlight that BA does not mimic nuclear DNA binding of Mitomycin (MC) for evoking non-selective, glucose-insensitive cell cycle arrest and caspase-3/7 apoptosis of T24 and TRT-HU1 cells. (HK2—hexokinase 2). Because faster proliferation of T24 cells amidst glucose scarcity is fueled by mitochondrial accumulation of fatty acids, hallmarked by faster glycogen depletion and lactic acid production (pH decline), the increased mitochondrial accumulation of a fatty acid mimetic (BA) inhibits fatty acid synthesis from glycogen, as indexed by slower glycogen decline amidst glucose scarcity. A mere 10% increase in glucose to 17.6 mM from a basal level of 16 mM for T24 doubled the IC_50_ of BA and BA-FITC relative to TRT-HU1 by reversing the upregulation of *caspase-3*, *p53*, *PTEN, GAPDH,* and *XIAP gene* expression induced by BA in T24 at basal glucose (16 mM). Thicker arrows signify higher uptake of BA and-FITC.

**Figure 2 ijms-26-05598-f002:**
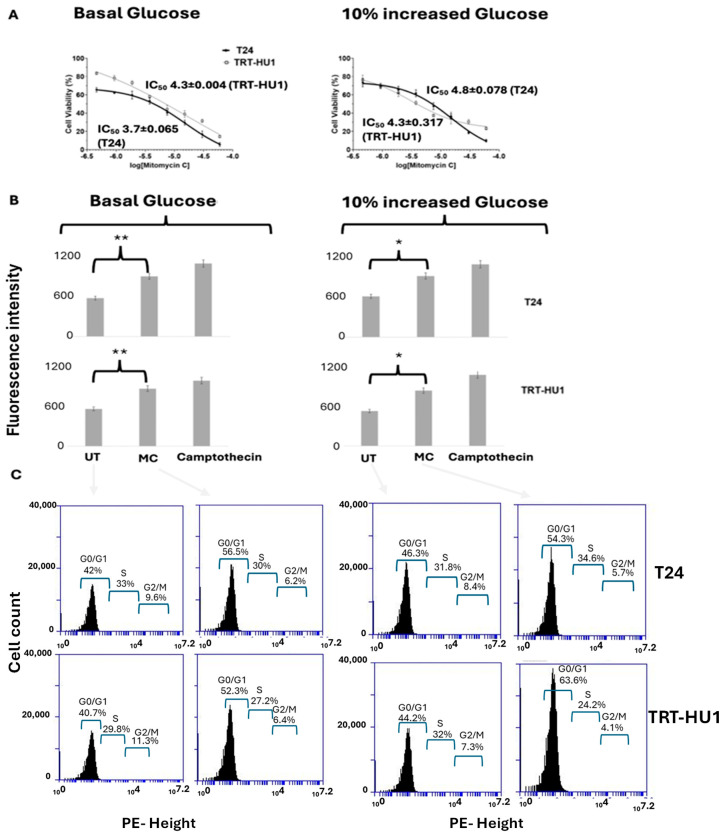
The standard chemotherapeutic drug, Mitomycin C (MC), exhibited non-selective cytotoxicity on carcinoma cell line T24 and benign urothelial cell TRT-HU1 at basal (**A**) as well as at 10% increased glucose from basal levels. The glucose-insensitive IC_50_ determined after 18 h exposure to MC is mirrored by caspase-3/7 apoptotic activity (**B**) at basal (** *p*-value ≤ 0.01) and at 10% increased glucose (* *p*-value ≤ 0.05) and cell cycle arrest at G0/G1 phase determined after 12 h exposure to MC (**C**).

**Figure 3 ijms-26-05598-f003:**
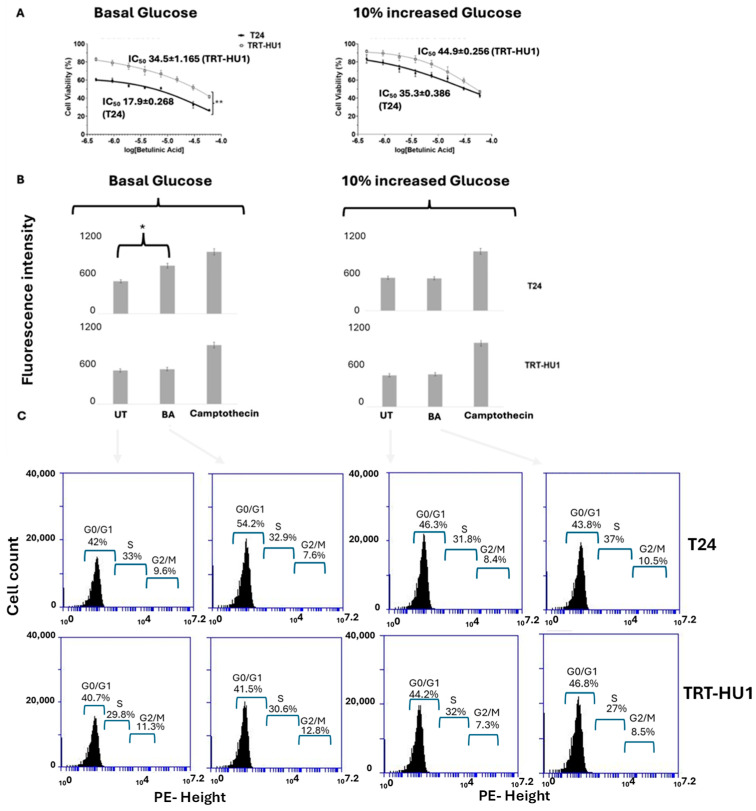
Betulinic Acid (BA) exhibited a significantly lower IC_50_ in T24 compared to TRT-HU1 grown at basal glucose (** *p* value ≤ 0.01 (**A**)), and the IC_50_ difference was abolished by a 10% increase in glucose from basal levels, which mirrored the results of the caspase-3/7 apoptotic assay (* *p* value ≤ 0.05 (**B**)). The doubling of BA IC_50_ with a 10% increase in glucose in T24 cells was determined by significantly lower apoptosis and lower cycle arrest as opposed to 13% higher cell cycle arrest (41.5% vs. 54.2%) in the G0/G1 phase of T24 cells grown at basal glucose (**C**). IC_50_ was determined by concentration-dependent antiproliferative action of BA after 18 h exposure, and cell cycle arrest and apoptosis were determined after 12 h exposure, using Camptothecin as a positive control in the caspase-3/7 apoptotic assay.

**Figure 4 ijms-26-05598-f004:**
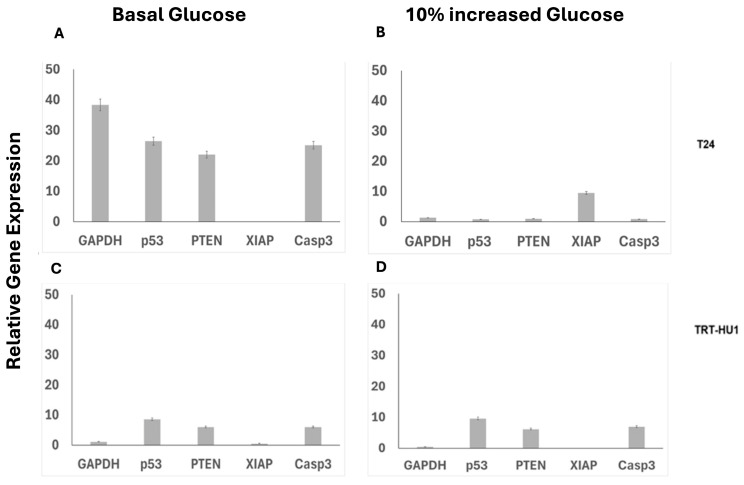
Lower IC_50_ of BA in T24 cells grown at basal glucose (**A**) is dependent on BA-induced up-regulation of pro-apoptotic genes: *Caspase-3* (Casp3) (*p*-value ≤ 0.01), *p53* (*p*-value ≤ 0.01), *PTEN* (*p*-value ≤ 0.01), and *GAPDH* (*p*-value ≤ 0.001), and the downregulation of the *XIAP* gene (*p*-value ≤ 0.05). While a 10% increase in glucose (**B**) from basal levels reversed BA-induced upregulation of *p53* and *Casp3* in T24 cells, TRT-HU1 cells were resistant to any BA-evoked gene alterations with or without an increase in glucose (**C**,**D**). Gene expression was calculated relative to untreated controls at basal and 10% increased glucose using 18S housekeeping gene.

**Figure 5 ijms-26-05598-f005:**
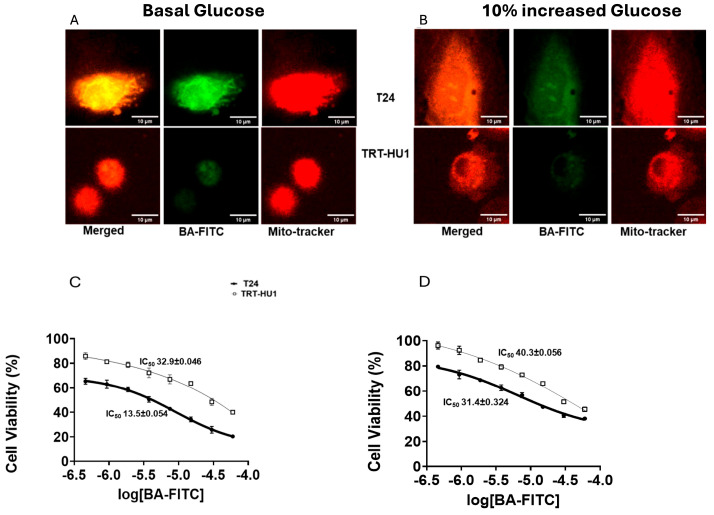
Glucose scarcity-dependent colocalization of the green fluorescence emitted by FITC conjugated to Betulinic Acid (BA-FITC; 20 µM for 2 h) with the red fluorescence emitted by Mito Tracker Red in T24 cells (**A**) is evident from a ~15% decline in Pearson’s correlation coefficient from 0.91 to 0.75 (graph shown in Appendix A) for fluorescence colocalization at basal glucose (**A**) and at 10% increased glucose (**B**), respectively, *p* value ≤ 0.01. The coefficient for colocalization, 0.66 and 0.63 in TRT-HU1 remained unchanged with a 10% increase in glucose from basal levels. Lower mitochondrial localization of BA-FITC in T24 at higher glucose resulted in a two-fold higher IC_50_ of BA-FITC (**D**) relative to the IC_50_ of BA-FITC at basal glucose (**C**). The cell viability values are normalized viability percentages relative to viability in the vehicle group.

**Figure 6 ijms-26-05598-f006:**
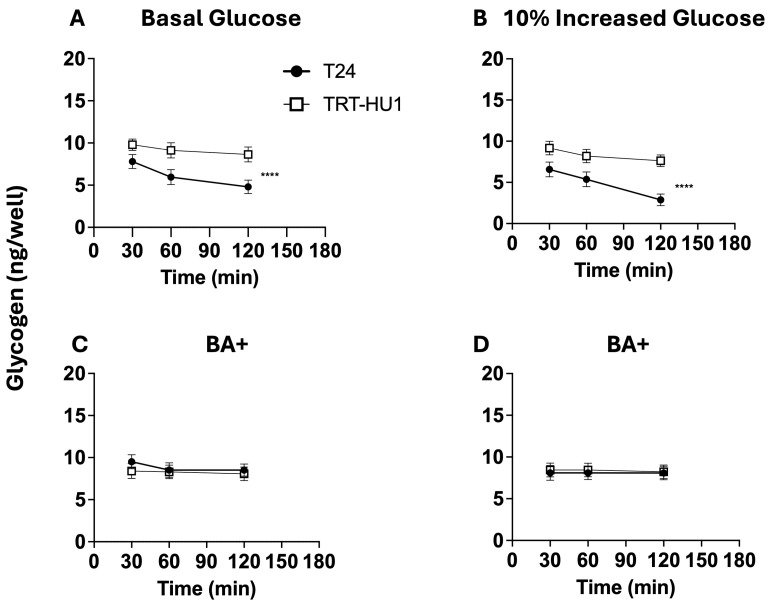
Glucose scarcity was evoked by replacing growth media containing basal or basal + 10% increased glucose with an unbuffered isotonic solution of pH 7.4. Steeper glycogen depletion (**A**,**B**) at 60 and 120 min (**** *p*-value < 0.0001) in the T24 cell line [•] than in TRT-HUI cells [□] at basal and basal + 10% increased glucose results from faster proliferation fueled by endogenous fatty acids synthesized from glycogen, and the enzymatic inhibition of that conversion by intracellular accumulation of fatty acid mimetic (BA) slows the glycogen depletion at basal (**C**) and at basal + 10% increased glucose (**D**). BA-evoked cell cycle arrest also slows the innate T24 proliferation amidst glucose scarcity.

**Figure 7 ijms-26-05598-f007:**
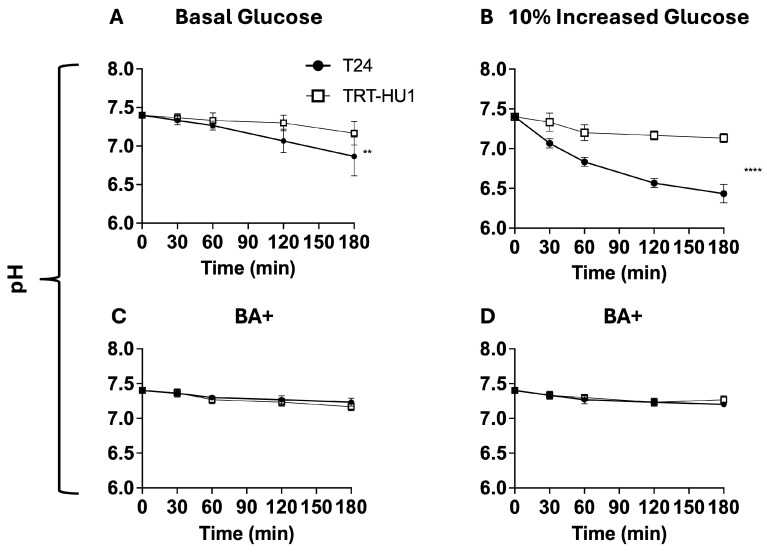
Glucose scarcity was evoked by replacing growth media containing basal or basal + 10% increased glucose with an unbuffered isotonic solution of pH 7.4. Glycogenolysis of glycogen stores followed by glycolysis fuels innately faster proliferation amidst glucose scarcity of the T24 cell line [∙] at basal (**A**) and +10% increased glucose (**B**) relative to TRT-HU1, which results in rapid extracellular acidification in a feed-forward manner, as displayed by steep pH decline at 60 and 180 min, ** *p* < 0.01 and **** *p* < 0.0001, respectively. A flatter slope of pH decline for BA-treated T24 and TRT-HU1 cells [□] grown at basal (**C**) and basal + 10% increased glucose (**D**) is consistent with reduced glycogenolysis and BA-evoked cell cycle arrest slowing the rate of extracellular acidification.

**Table 1 ijms-26-05598-t001:** Two-fold faster doubling time of T24 than TRT-HU1 is dependent on five-fold higher basal glucose levels of 16 mM and 3 mM in McCoy’s 5A growth media and Keratinocyte Serum Free media, respectively. An additional 10% glucose increase from basal levels to 17.6 mM and 3.3 mM significantly decreased the doubling time by >2.8 h vs. ~1 h for T24 and TRT-HU1, respectively.

Doubling Time (h)	T24	TRT-HU1
Basal Glucose	18.3 ± 0.048	40.7 ± 0.024
10% Increased Glucose	15.5 ± 0.126	39.4 ± 0.072

## Data Availability

Appendix A can be found at the University of Pittsburgh’s shared drive folder.

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
