# Peer review of "Metabolic Reprogramming of Urothelial Carcinoma—A Theragnostic Target for Betulinic Acid"

_ijms, 2025, doi:10.3390/ijms26125598_

Round 1
Reviewer 1 Report
Comments and Suggestions for Authors
1) The physiological relevance of using a 10% increase in glucose concentration to mimic hyperglycemia is unclear. A 10% increase in glucose corresponds roughly to blood glucose levels seen in conditions such as prediabetes or mild diabetes. Please provide specific references or clinical evidence demonstrating that this mild increase in glucose is relevant and sufficient to influence tumor biology or drug sensitivity in urothelial carcinoma patients.
2) Mitomycin C is commonly utilized clinically for the treatment of superficial (non-muscle-invasive) bladder cancer, rather than muscle-invasive bladder cancer represented by the T24 cell line used in this study. Thus, Mitomycin C may not be an appropriate comparative control for evaluating the therapeutic efficacy of Betulinic Acid (BA) in invasive urothelial carcinoma models. Cisplatin, a standard chemotherapy drug routinely used in treating muscle-invasive bladder cancer, would serve as a more suitable control. Consider either replacing Mitomycin C with Cisplatin or clearly justifying the rationale for selecting Mitomycin C as the comparator in this context.
Author Response
1) The physiological relevance of using a 10% increase in glucose concentration to mimic hyperglycemia is unclear. A 10% increase in glucose corresponds roughly to blood glucose levels seen in conditions such as prediabetes or mild diabetes. Please provide specific references or clinical evidence demonstrating that this mild increase in glucose is relevant and sufficient to influence tumor biology or drug sensitivity in urothelial carcinoma patients.
Response: We thank the reviewer for highlighting a clinically relevant point and appreciate the opportunity to clarify here. We selected 10% increase from basal glucose levels to ensure physiological relevance for our findings because even with a 10% increase, the glucose levels of 3mM[ 52.5mg/ 100mL (dL)] in media for non-neoplastic cell line TRT-HU1 fall below the lower threshold for the normal human blood glucose range of 3.9-7mM whereas basal glucose levels of 16mM (280mg/ dL) in commercially available McCoy’s 5A media for neoplastic cell line T24 correspond to sustained hyperglycemia of a severely diabetic person having glycosylated hemoglobin levels, HbA1C ≥11. This correspondence between basal glucose levels of 16mM in McCoy’s 5A media and HbA1C sheds light on the association between HbA1C ≥ 7 and significantly higher recurrence rate and worse recurrence-free survival of non-muscle invasive bladder cancer patients in a retrospective study (new ref. 10. PMID: 32758196). The increased risk of urothelial carcinoma in diabetic individuals is substantiated by a meta-analysis (ref. 11 and 12). In addition, hyperglycemia as an adverse effect of therapy (PMID: 37678672) influenced the tumor biology and lowered the drug sensitivity of antibody drug conjugates, Enfortumab vedotin in 6.8% of urothelial carcinoma patients (reference 10 of the manuscript) and high glucose levels reduce the uptake of the probe, 2- Fluoro-2-deoxy-D-glucose (2-FDG) in cancer cells (PMID: 9074524; 31304560), which spoils the tumor signal in PET scan. Importantly, the lower sensitivity of Enfortumab vedotin in hyperglycemic bladder cancer patients substantiates the lower sensitivity of Betulinic acid and Betulinic acid conjugated FITC on neoplastic T24 cell line grown in McCoy’s 5A media with 10% increased glucose relative to the IC50 of Betulinic acid and Betulinic acid conjugated FITC measured at basal levels of 16mM in McCoy’s 5A media. In contrast to the glucose sensitive anticancer effect of Enfortumab vedotin or Betulinic acid, the cytotoxic effect of mitomycin on T24 cell line was insensitive to the 10% increase in glucose from basal levels.
Considering the correlation between blood glucose levels and the glucose levels of cell growth media, one can easily deduce that commercially available McCoy’s 5A media recapitulates the glucose levels of a severe diabetics to fuel two-fold faster proliferation of T24 cell line than non-neoplastic cell line, TRT-HU1. Because blood glucose levels of 25mM (450mg/dL) do not represent the majority population of bladder cancer patients, our study raises sharp questions on the clinically relevant rise in glucose levels of T24 cells for studying biology of urothelial carcinoma, whether it should be >50% (PMID: 32678516) or a 10% increase from basal levels of 16mM Glucose in McCoy’s 5A media. Based on our findings, we fear that T24 grown in McCoy’s 5A media may be underestimating the in vivo efficacy of anticancer drugs such as Betulinic drugs that do not alkylate DNA like Mitomycin or target metabolic reprogramming of cancer. Overall, a large body of clinical and epidemiological evidence establishes a 10% increase in normal blood glucose (hyperglycemia) and aberrant glucose metabolism as an independent risk factor for increased rates of cancer and death (PMID: 32547697, 33562380) and we have now revised the manuscript from line 44 onwards to address this comment.
2) Mitomycin C is commonly utilized clinically for the treatment of superficial (non-muscle-invasive) bladder cancer, rather than muscle-invasive bladder cancer represented by the T24 cell line used in this study. Thus, Mitomycin C may not be an appropriate comparative control for evaluating the therapeutic efficacy of Betulinic Acid (BA) in invasive urothelial carcinoma models. Cisplatin, a standard chemotherapy drug routinely used in treating muscle-invasive bladder cancer, would serve as a more suitable control. Consider either replacing Mitomycin C with Cisplatin or clearly justifying the rationale for selecting Mitomycin C as the comparator in this context. We address this comment in the revised discussion of the manuscript at line 270.
Response: We thank the reviewer for the critical analysis of our findings from a clinical perspective. We would like to submit that Mitomycin C is routinely used as an active comparator for evaluating the efficacy of new bladder cancer drugs on T24 cell line and use of T24 and mitomycin as search terms generated 80 pubmed results. Because the cytotoxicity of Betulinic acid on T24 cell line was inferior to Mitomycin C, one can easily predict that Betulinic acid will be also inferior to Cisplatin in T24 cell line. However, the goal of our research was not to generate evidence for Betulinic acid to be a potential substitute for Mitomycin or Cisplatin in the treatment of non-muscle invasive cancer or in muscle invasive cancer but to report the mechanistic evidence that underpins the cancer cell selectivity (T24 vs TRT-HU1) of BA and conjugates compared to Mitomycin. Unlike the comparable IC50 of Mitomycin on neoplastic and non-neoplastic cell lines, Betulinic acid was eight times more toxic on T24 as well as RT4 (a non-muscle invasive bladder cancer cell line) than on non-neoplastic cell lines of TRT-HU1 and HBDEC. We also determined that metabolic reprogramming of cancer to ensure proliferation amidst glucose scarcity is a key determinant in the elevated toxicity of BA on T24 cells and this elevated BA toxicity of T24 cells at basal glucose is abolished by raising glucose levels. These findings underpin the projected use of BA conjugates as potential theragnostic to detect and surveil bladder cancer patients.
Reviewer 2 Report
Comments and Suggestions for Authors
The authors conducted a preclinical study assessing the impact of glucose metabolism on urothelial cell lines.
According to MDPI guidelines, experiments should include at least two different cancer cell lines. However, the current study employs only one cancer cell line and one non-neoplastic human bladder cell line. To clearly distinguish the role of urothelial carcinoma from other confounding factors inherent to the neoplastic nature of the chosen cell line, an additional cancer cell line should be introduced.
In the dose-response curves presented (Figures 3 and 5), the neoplastic and non-neoplastic cell lines exhibit differing percentages of cell viability even at very low doses of BA or BA FITC. It would be beneficial if the authors provided normalized viability percentages relative to the initial conditions. Adjusting the data in this way might reduce observed differences and impact the calculated IC50 values.
Additionally, the manuscript lacks mechanistic evidence connecting BA FITC and mitochondrial co-localization. Given that mitochondrial membranes can become disrupted during apoptosis, affecting co-localization measurements, no definitive conclusion regarding the direct mitochondrial activity of the BA FITC conjugate can be drawn from the presented experiments. It is possible that the observed mitochondrial accumulation is an artifact of normal conditions rather than a direct effect of the compound.
Furthermore, exploring the resistant phenotype is essential and currently missing from the manuscript. Investigating resistance could significantly enhance the understanding of the mechanism underlying the action of the BA FITC conjugate.
Lastly, Seahorse metabolic assays should be performed to evaluate metabolic changes at baseline conditions and following treatments with BA and BA FITC to robustly validate alterations in metabolism.
Comments on the Quality of English Language
The manuscript is difficult to follow in many passages.
The introduction should be rephrased: i)state of art, ii) gap in literature, iii) research question.
Author Response
We thank the reviewers for constructive comments to our manuscript, which help make the manuscript stronger. The changes in manuscript are highlighted in yellow and the response to specific comments is stated below
- The authors conducted a preclinical study assessing the impact of glucose metabolism on urothelial cell lines.According to MDPI guidelines, experiments should include at least two different cancer cell lines. However, the current study employs only one cancer cell line and one non-neoplastic human bladder cell line. To clearly distinguish the role of urothelial carcinoma from other confounding factors inherent to the neoplastic nature of the chosen cell line, an additional cancer cell line should be introduced.
Response: We thank the reviewer for highlighting the elements of scientific rigor on the number of cell lines. Indeed, we studied the effect of Betulinic acid on four cell lines, two neoplastic (T24 and RT4) and two non-neoplastic (TRT-HU1 and HBDEC). While the main manuscript is focused on T24 and TRT-HUI cell lines, supplementary data includes data on two other cell lines. Therefore, based on the anticancer effect of Betulinic acid in T24 and RT4 cell line, we can distinguish the role of urothelial carcinoma from other confounding factors inherent to the neoplastic nature of the chosen cell line. Moreover, prior study of the first author had studied the anticancer effect of Betulinic acid (BA) and Bis-arylidene oxindole-betulinic Acid conjugates and BA-FITC on a host of cell lines (cited reference 14 of the manuscript). This comment is now addressed in the revised introduction at line 94.
2)In the dose-response curves presented (Figures 3 and 5), the neoplastic and non-neoplastic cell lines exhibit differing percentages of cell viability even at very low doses of BA or BA FITC. It would be beneficial if the authors provided normalized viability percentages relative to the initial conditions. Adjusting the data in this way might reduce observed differences and impact the calculated IC50 values.
Response: We thank the reviewer for the critical review of data in Figure 3 and 5. We would like to clarify that the viability percentages were calculated relative to the average of the vehicle control wells. While a random variation is not discernible in the viability% at the lowest dose of mitomycin (Fig 2A) and of BA (Fig.3A) for T24 cells grown in basal glucose, variation is discernible for lowest dose of BA-FITC in T24 cells (Fig 5C) grown at basal glucose, and the variation gets diminished for all the tested agents when T24 cells are grown at 10% higher glucose from basal levels. This random variation in T24 cell viability% at the lowest dose of BA and BA-FITC is fully accounted in the fitting of sigmoidal dose-response curve to a non-linear regression model by Graphpad prism for determining IC50, a mean response that is 50% of the lowest dose. Accordingly, despite a slightly lower viability % of T24 cells (basal glucose) at the lowest dose of BA than for BA-FITC, the IC50 of BA remains higher than for BA-FITC, 17.9 mM vs 13.5mM and this ~4mM difference in IC50 is upheld even with elevated IC50 of BA and BA-FITC, 35.3mM vs 31.4mM for T24 cells grown at 10% higher glucose which ensures almost identical viability at the lowest dose of BA(Fig.3A) and BA-FITC (Figure 5D).Overall, our findings attest to the robustness of IC50 as measure of efficacy and its independence from the slight variation in initial viability is also reported by others(PMID: 18582601; 26320443). To further check the reviewer concern, we also calculated IC50 after plotting the viability percentages relative to the initial condition of lowest dose in Figure 5C and D and those values are plotted in supplementary data figure S7. The IC50 of BA-FITC at basal glucose in T24 cell remained the same. This comment is now addressed in the revised introduction at line 229.
3) Additionally, the manuscript lacks mechanistic evidence connecting BA FITC and mitochondrial co-localization. Given that mitochondrial membranes can become disrupted during apoptosis, affecting co-localization measurements, no definitive conclusion regarding the direct mitochondrial activity of the BA FITC conjugate can be drawn from the presented experiments. It is possible that the observed mitochondrial accumulation is an artifact of normal conditions rather than a direct effect of the compound.
Response: We thank the reviewer for an insightful comment which helped us draw a key insight into the mechanistic basis for higher toxicity (lower IC50) of Betulinic acid in non-neoplastic cell lines TRT-HU1 (Figure 3) and HBDEC cell line (Figure S5B) grown at 10% higher glucose than the IC50 of Betulinic acid measured on TRT-HU1 and HBDEC grown at basal glucose. The concern of artifact in mitochondrial accumulation is contradicted by a large body of published evidence (PMID: 9852046; 33806566) and by neoplastic (T24) and non-neoplastic cells (TRT-HU1) left untreated with BA-FITC (Supplementary figure 1) displaying comparable intensity of Mito Tracker red fluorescence to cells receiving sequential treatment of Mito Tracker red and BA-FITC (Figure 5). Therefore, it is logical to infer that BA-FITC could be tracing the pathway paved by Mito Tracker red to enter mitochondria and Betulinic acid may also be relying on mitochondrial membrane transformations (PMID: 32931774) for cancer selectivity. Moreover, higher mitochondrial accumulation MitoTracker red in non-neoplastic cells (TRT-HU1) grown at 10% higher glucose implies that glucose toxicity of non-neoplastic TRT-HU1 cells replicate mitochondrial membrane transformations seen in neoplastic cells (T24 cells) grown at basal glucose. The differential in Mito tracker red uptake between neoplastic and non-neoplastic cells at basal and 10% higher glucose appears to be linked to the difference in their respective doubling times. Higher MitoTracker red localization in the mitochondria of slow dividing TRT-HU1 at 10% higher glucose indexes that higher intracellular levels of glucose increase the mitochondrial uptake of Mito tracker red whereas the “Warburg effect” of fast dividing T24 cell line elucidate the decline in mitochondrial uptake of Mito tracker red with rise in glucose from basal levels (PMID: 30108714). Moreover, the mitochondrial accumulation of Betulinic acid in T24 and TRT-HU1 cells is consistent with the cell cycle arrest G0/G1 phase and induction of cell death via apoptosis (Figure 3) and glycogen stores and pH decline (Figure 6 and 7), respectively. This response is addressed on page 8 and 9 of the manuscript.
4)Furthermore, exploring the resistant phenotype is essential and currently missing from the manuscript. Investigating resistance could significantly enhance the understanding of the mechanism underlying the action of the BA FITC conjugate.
Response: We thank the reviewer for this suggestion. In the present manuscript, we were focused on dissecting the mechanistic basis for the elevated toxicity of Betulinic acid in neoplastic cell lines relative to non-neoplastic cell lines. The reported evidence makes it abundantly clear that metabolic reprogramming of cancer to proliferate amidst glucose scarcity is a determinant for cancer cell selectivity of Betulinic acid. Moreover, as also discussed in the response to reviewer 1’s first comment, hyperglycemia is a well-recognized determinant for the resistant bladder cancer phenotype (PMID: 32758196; PMID: 37678672) and a 10% rise in glucose makes neoplastic cell lines (T24 and RT4) resistant to the anticancer effect of Betulinic acid. By raising the glucose levels by 10% from the basal levels in growth media, we also determined that T24 cells mirror the glycogen decline seen in cancer cells isolated from the urine of bladder cancer patients (PMID: 28541036) and glycogen decline is correlated with the internalization of BA and BA-FITC conjugates, which is a prerequisite for targeting metabolic reprogramming of bladder cancer cells. Moreover, glucose toxicity of non-neoplastic urothelial cells (TRT-HU1 and HBDEC) accentuated the cytotoxicity of Betulinic acid. This response is addressed in the discussion section, page 8 of the manuscript.
5) Lastly, Seahorse metabolic assays should be performed to evaluate metabolic changes at baseline conditions and following treatments with BA and BA FITC to robustly validate alterations in metabolism.
Response: We thank the reviewer for this suggestion. As stated in the previous response, our primary aim in this manuscript was to report the evidence generated from the testing of the null hypothesis that a 10% rise in glucose from respective basal levels does not impact the elevated cytotoxicity of Betulinic acid in neoplastic cell lines of T24 and RT4 compared to non-neoplastic cell lines TRT-HU1 and HBDEC. Based on the evidence from control experiments of non-neoplastic cells reported in the manuscript, we can reject the null hypothesis with reasonable confidence. In addition, we showed that a 10% rise in glucose from basal levels impacts glycogen depletion, extracellular acidification (pH decline) as surrogate for glycolysis and gene expression together with cell cycle arrest and apoptosis evoked by Betulinic acid. We agree that future manuscripts of further studies will robustly validate the metabolic alterations evoked by BA and BA-FITC in future steps of this project using Seahorse metabolic assay to simultaneously measure extracellular acidification with oxygen consumption in neoplastic and non-neoplastic cells. However, Seahorse metabolic assay is outside the scope of this manuscript and does not impact the main conclusion that glucose scarcity driven metabolic reprogramming of cancer to proliferate is central to the cancer cell selectivity of Betulinic acid and its conjugates. This response is addressed in the discussion section, page 11, line 373 of the manuscript.
6) The introduction should be rephrased: i)state of art, ii) gap in literature, iii) research question.
Response: We thank the reviewer for this constructive suggestion, and we have rephrased the introduction in that manner.
Round 2
Reviewer 2 Report
Comments and Suggestions for Authors
The authors addressed my concerns. No other comments
Author Response
We thank the reviewers for constructive feedback to our manuscript. The revised manuscript address these comments in the discussion.
Comment 1.Please provide more context on the potential clinical relevance of the findings, specifically how the glucose sensitivity of BA could be translated into therapeutic applications for urothelial carcinoma.
Response: We thank the reviewer for highlighting an important point, which we have addressed in the discussion on page 12 of revised manuscript.
2. What are the potential challenges or limitations in using BA as a therapeutic agent for bladder cancer in patients with hyperglycemia, and how can these be addressed in future clinical applications?
Response: We thank the reviewer for raising this point and the manuscript already dealt with this in the discussion on page 12.
2. What are the potential challenges or limitations in using BA as a therapeutic agent for bladder cancer in patients with hyperglycemia, and how can these be addressed in future clinical applications?
Response: We thank the reviewer for raising this point and the manuscript already dealt with this in the discussion on page 12 of the manuscript.